# Pathways to reduced physical intimate partner violence among women in north-western Tanzania: Evidence from two cluster randomised trials of the MAISHA intervention

Tanya Abramsky[1]*, Diana Sanchez Guadarrama[1], Saidi Kapiga[2,3], Grace Mtolela[2], Flora Madaha[2], Shelley Lees[1], Sheila Harvey[1]

1 Department of Global Health and Development, London School of Hygiene and Tropical Medicine, London, United Kingdom, 2 Mwanza Intervention Trials Unit, Mwanza, Tanzania, 3 Department of Infectious Diseases Epidemiology, London School of Hygiene and Tropical Medicine, London, United Kingdom

* Tanya.abramsky@lshtm.ac.uk

**Data Availability Statement:** The MAISHA study datasets are available, upon request, to members

## Abstract

Intimate partner violence (IPV) affects over one-in-four women globally. Combined economic and social empowerment interventions are a promising IPV prevention model. However, questions remain on the mechanisms through which such interventions prevent IPV, and whether standalone social empowerment interventions can work in the absence of an economic component. This secondary analysis of MAISHA Study data (north-western Tanzania) explores pathways through which a group-based gender-training intervention, delivered to women standalone or alongside microfinance, may impact on physical IPV risk. Two cluster-randomised trials (CRT) assessed the impact of the MAISHA intervention on women's IPV risk; CRT01 among women in 66 pre-existing microfinance groups (n = 919), and CRT02 among 66 newly-formed groups not receiving microfinance (n = 1125). Women were surveyed at baseline and 29 months follow-up. Sub-group analyses explored whether intervention effects on past-year experience of physical IPV varied by participant characteristics. Mediators of intervention effect on physical IPV were explored using mixed-effects logistic regression (disaggregated by trial). In CRT01, MAISHA was associated with reduced past-year physical IPV (adjusted-OR 0.63, 95%CI 0.41–0.98), with stronger effects among those younger, more financially independent, and without prior physical IPV. CRT02 showed no impact on physical IPV, overall or among sub-groups. In CRT01, individual-level reduced acceptability of IPV and group-level confidence to intervene against IPV emerged as potential mediators of intervention effect, while relationship-level indicators of communication were not impacted. In CRT02, positive impacts on individual-level attitudes did not translate into reduced IPV risk. In CRT02, arguments with partners over perceived transgressions of gender roles increased in the intervention-arm. Neither trial resulted in increased separations. Findings illustrate the importance of addressing poverty and women's economic dependence on men, structural factors that may impede the success of socially oriented violence prevention

of the scientific and medical community for non-commercial use only. These datasets are stored in Data Compass, the London School of Hygiene and Tropical Medicine's digital data repository and can be requested via the following DOIs: https://doi.org/10.17037/DATA.00003591 (CRT01) and https://doi.org/10.17037/DATA.00003592 (CRT02). Researchers may apply for access to data on condition that they comply with the ethical obligations and consent conditions of the original study (for reference, informed consent documents can be accessed using the above links). Data requests that are submitted through the LSHTM Data Compass are automatically sent to the study team and LSHTM Research Data Management Service (a non-author contact) for follow-up. For additional queries, the LSHTM Research Data Management Service can be contacted by emailing researchdatamanagement@lshtm.ac.uk.

**Funding:** The study was funded by: an anonymous donor (TA, SK, GM, FM, SL, SH); and the STRIVE Research Consortium which was funded by UK Aid from the UK Department for International Development (SK, GM, FM SL SH - grant number PO 5244 , https://www.ukaiddirect.org/). An additional grant was provided by the anonymous donor for secondary analysis of the MAISHA data (TA, SH, SL, SK). The funders had no role in study design, data collection and analysis, decision to publish, or preparation of the manuscript.

**Competing interests:** The authors have declared that no competing interests exist.

programming. Programming with men is also crucial to ameliorate risks of backlash against attitudinal/behavioural change among women.

**Trial registration:** ClinicalTrials.gov #NCT02592252.

## Introduction

Globally, over a quarter of woman have experienced physical and/or sexual violence by a partner during their lifetime [1]. This violence can have serious consequences for women's physical and mental health, the health and wellbeing of their children, and their participation in social and economic activity [2–4].

Recent years have seen a growth in research on violence against women (VAW), with the evidence base now showing that VAW is preventable through a range of interventions. One model that has been shown to be effective in multiple well-conducted evaluations is combined economic and social empowerment interventions for women [5–8].

The IMAGE intervention in South Africa, combining microfinance and group-based gender training sessions for women, was the first such intervention shown to reduce women's experience of physical and sexual intimate partner violence (IPV) in a cluster randomised trial [9]. Since then, evidence from other contexts has supported the potential for similar interventions to reduce IPV [10, 11].

The rationale for interventions that combine economic and social empowerment components has long been that bigger and broader impacts can be achieved by simultaneously tackling multiple strands of structural disadvantage faced by women [9, 12]. At the same time as challenging inequitable gender attitudes and norms, building women's confidence and fostering communication and conflict resolution skills, they also address women's economic dependence on men. They thereby enhance women's power *and* resources to be more discriminate in partner choice, negotiate the parameters of new or existing relationships, or leave abusive relationships [13]. There is now mounting evidence that economic interventions combined with gender transformative interventions are more consistently associated with reductions in IPV than are economic interventions alone [5, 14, 15]. The question, however, of whether gender transformative interventions alone can prevent IPV remains under-researched. Are gender training interventions sufficient in their own right to reduce IPV in contexts where women have limited access to and control over economic resources?

The MAISHA study, comprising two cluster randomised trials of a 10-session gender training intervention to prevent IPV, offers an opportunity to explore this question—one trial conducted among women in pre-established microfinance groups [16] and the other among women not receiving microfinance in Mwanza city, Tanzania [17]. When delivered to women in established microfinance groups (CRT01), the MAISHA intervention was associated with a reduction in women's past-year experience of physical IPV (aOR 0.63, 95% CI 0.41–0.98) [18]. The same intervention, delivered to women in newly formed neighbourhood groups not receiving microfinance (CRT02) had no impact on past-year physical IPV (aOR 0.98, 95%CI 0.72–1.33) [19]. Attendance at MAISHA sessions was high in both trials, particularly so in CRT02 (82% attended 7 or more of the 10 sessions in CRT02, versus 67% in CRT01) [18, 19]. The question thus arises as to why the same gender training intervention led to a reduction in past-year physical IPV among women in pre-established microfinance groups, but not among women in newly formed groups not receiving microfinance.

This paper attempts to unpick how and among whom the MAISHA intervention worked to prevent physical IPV among women in pre-established microfinance groups (CRT01). We

then seek to assess the extent to which those same pathways were present, absent or modified among women in CRT02.

First, we explore whether intervention effectiveness in each trial varied between different sub-groups of women. We discuss whether any observed differential effects between sub-groups in CRT01 might explain the null impacts on physical IPV in CRT02 given differences in the underlying characteristics of women enrolled in the two trials. Second, we identify potential mediators through which MAISHA may have impacted on physical IPV in CRT01, and assess alternative explanations as to why the intervention did not reduce past year physical IPV in CRT02: a) the relevant mediators didn't change; b) mediators changed but did not translate into reductions in IPV; c) certain mediators changed in an unintended direction.

## Materials and methods

### Study setting and study design

The MAISHA study, comprising two cluster randomised trials (CRT01 and CRT02) was conducted in Mwanza city in north-western Tanzania. IPV is very common in Tanzania; overall, 44% of ever-married women report having experienced physical or sexual violence from their husband or partner [20]. Patriarchal norms are prevalent, over half of women and 40% of men reporting attitudes accepting of a man's use of violence against his wife [20]. Drivers of IPV also include early marriage, with more than a third of women married before the age of 18. Young brides are more likely to drop out of school and to begin childbearing early, resulting in limited economic opportunities and subsequent economic dependence on male partners [21]. Overall, among women (aged 15–49 years), 62% report being currently married or living together, while 10% report being divorced or separated [20].

The MAISHA study was implemented by the Mwanza Intervention Trials Unit, the Tanzania National Institute for Medical Research, and the London School of Hygiene & Tropical Medicine in close collaboration with local leaders and community members. The trials are described in full elsewhere [18, 19].

The CRT01 trial recruited 66 existing microfinance loan groups located in three wards in Mwanza city. Of these, 33 groups (544 women) were randomised to receive the MAISHA gender training intervention, and 33 groups (505 women) were waitlisted to receive the intervention after trial completion (control). The CRT02 trial was also conducted in Mwanza city but in different wards where BRAC (the microfinance provider in CRT01) was known to be less active. The trial team worked in collaboration with local leaders to form 66 neighbourhood groups of women who were not engaged in a formal microfinance loan scheme. Of these, 33 groups (627 women) were randomised to receive the MAISHA gender training intervention, and 33 groups (638 women) were waitlisted (control). Details of the randomisation process are described elsewhere [16, 17]. CONSORT flow diagrams for the two trials are presented in Fig 1 (CRT01) and Fig 2 (CRT02).

### CRT01—Microfinance groups

BRAC, an established microfinance provider in Tanzania, provides microfinance loans to women of low socioeconomic status with no access to formal financial services. Women are organised into groups that meet every week to repay part of their loan. Established microfinance groups were eligible for inclusion in the MAISHA CRT01 trial if they had fewer than 30 active members and a good meeting attendance/repayment record. A microfinance loan group was only enrolled if at least 70% of members provided written informed consent, and only those individuals who consented participated in trial activities. The recruitment process is described in detail elsewhere [16].

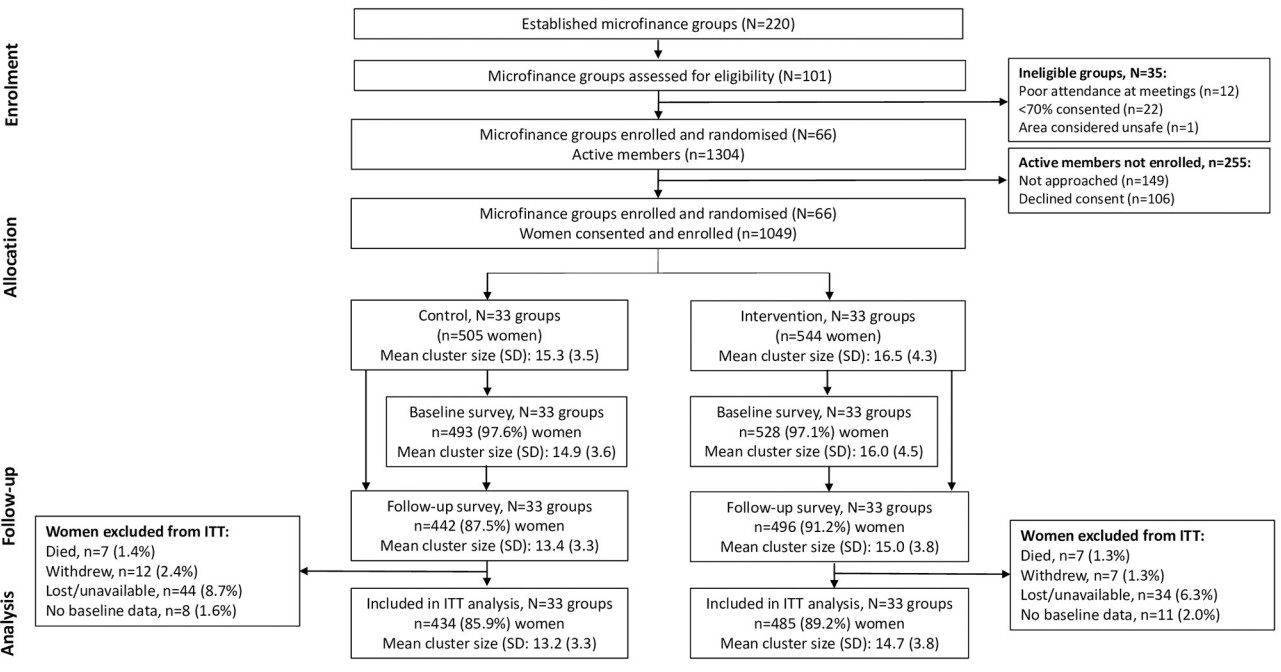

**Fig 1. CONSORT flow diagram for CRT01.**

The microfinance loan scheme was implemented by BRAC independently of the research team, and, as per usual BRAC procedure, women in both arms of the trial met weekly for loan repayments. On alternate weeks, either before or after the loan group meeting, groups allocated to the intervention arm participated in the 10-session MAISHA intervention.

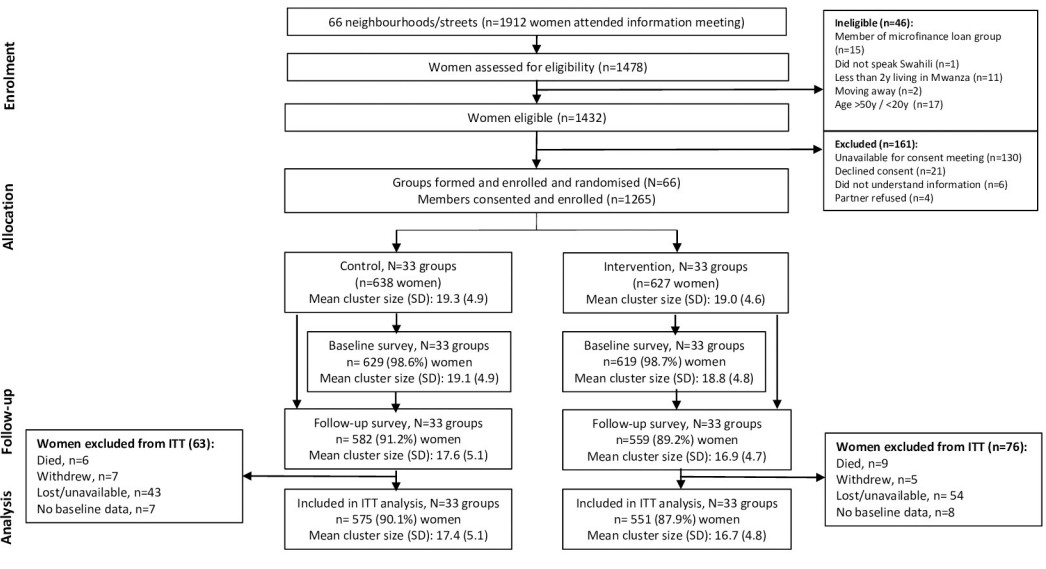

**Fig 2. CONSORT flow diagram for CRT02.**

## CRT02—Neighbourhood groups

For CRT02, the trial team worked with community leaders to identify neighbourhoods that were similar, in terms of levels of urbanisation, to those in which women in CRT01 lived (but in which BRAC were less active). Once a potential neighbourhood had been identified, MAISHA team members worked with local leaders to identify and visit households in the neighbourhood in order to invite potentially eligible women to attend information meetings [17]. Women were eligible for inclusion if they were aged 20–50 years, resident in Mwanza for 2 years or more, not a member of a formal microfinance loan group scheme in the past 12 months, fluent in Swahili, and not in formal employment. Women who demonstrated comprehension of the trial procedures and agreed to take part were invited to sign the consent form. Within each neighbourhood, the trial team aimed to recruit approximately 15–20 women.

## The MAISHA intervention

The MAISHA curriculum was developed by EngenderHealth (an international non-profit organisation focusing on gender equity and reproductive health) in collaboration with the research team. It draws upon other published curricula, including Sisters for Life from the IMAGE study [9], and was designed to be participatory and reflective. The MAISHA intervention aimed to empower women, prevent IPV, and promote healthy relationships by increasing women's knowledge and awareness (eg, of the consequences of normative attitudes to gender and IPV), developing their relationship skills (eg, communication and conflict resolution), and improving their social capital and peer support networks.

In both the CRT01 and CRT02 trials, the 10-session MAISHA intervention (outlined in Fig 3) was delivered to the 33 intervention arm groups on alternate weeks over a 20-week period. Each session lasted between 1.5 and 2 hours. Venues were selected to be convenient to participants, with sessions generally taking place at the group leader's house or in a quiet area of a local café or guesthouse. Sessions were delivered by trained female facilitators, following the *Wanawake Na Maisha* curriculum. Facilitators were recruited and trained by the research team and EngenderHealth on curriculum content, materials and facilitation skills. Intervention delivery was monitored by the trial coordinator and senior research team members.

## Data collection

In both trials, data were collected at two time-points, the baseline survey conducted prior to randomisation, and the follow-up survey conducted 29 months post randomisation (24 months after the intervention groups had completed the intervention sessions). Data collection took place between 2014 and 2018 for CRT01, and 2015 and 2019 for CRT02.

The baseline and follow-up surveys comprised structured questionnaires, covering respondents' sociodemographic characteristics, attitudes and norms relating to gender and IPV, partner characteristics and relationship dynamics, childhood exposure to violence, and participation in community groups and activities. Violence questions were taken from the WHO Violence Against Women instrument [22], which has also been widely used in Demographic and Health Surveys [23] and other trials of IPV prevention interventions [11, 24–26].

The questionnaire was developed in English, translated into the local language (Swahili) and independently back-translated into English for validation. Interviews were conducted face-to-face in private by female interviewers trained in interviewing techniques, gender issues, VAW, and ethical issues related to IPV research.

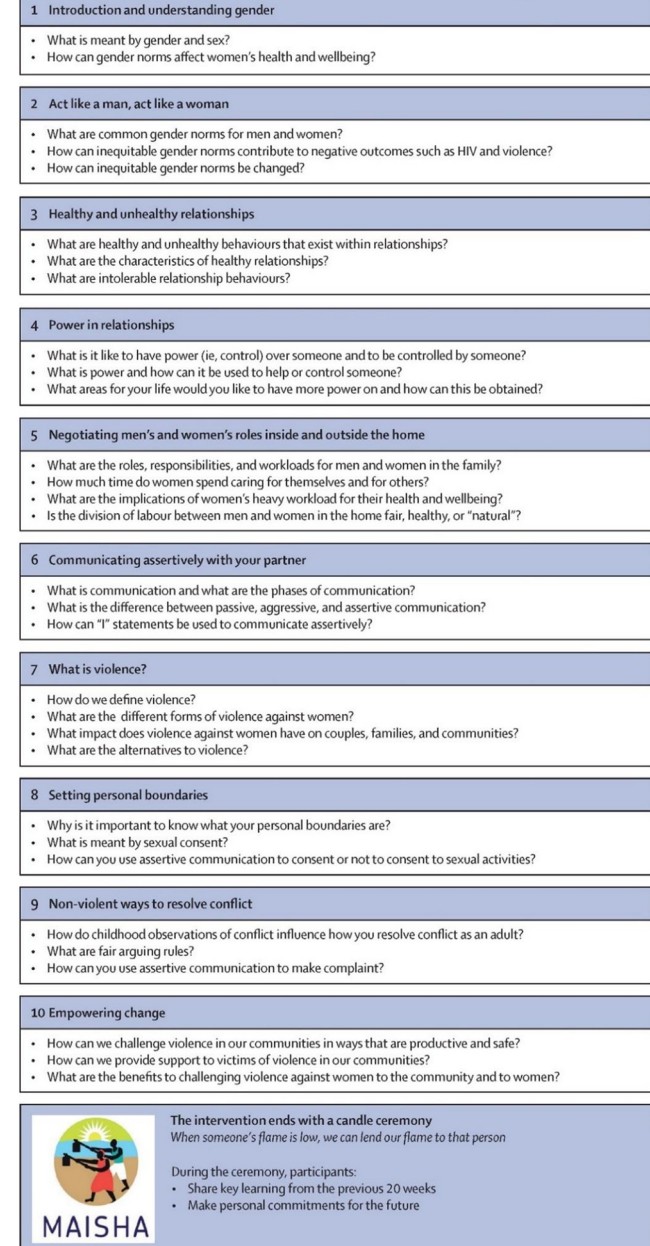

**Fig 3. Outline of the Wanawake Na Maisha curriculum.**

## Conceptual framework

The pathways through which the MAISHA intervention is hypothesised to impact on physical IPV, and participant characteristics that might modify impact, are laid out in Fig 4. This conceptual framework was developed based on the MAISHA theory of change (S1 Fig), insights from the MAISHA qualitative data [27], broader IPV risk factor and prevention literature [6] and the availability of data items from the study questionnaires.

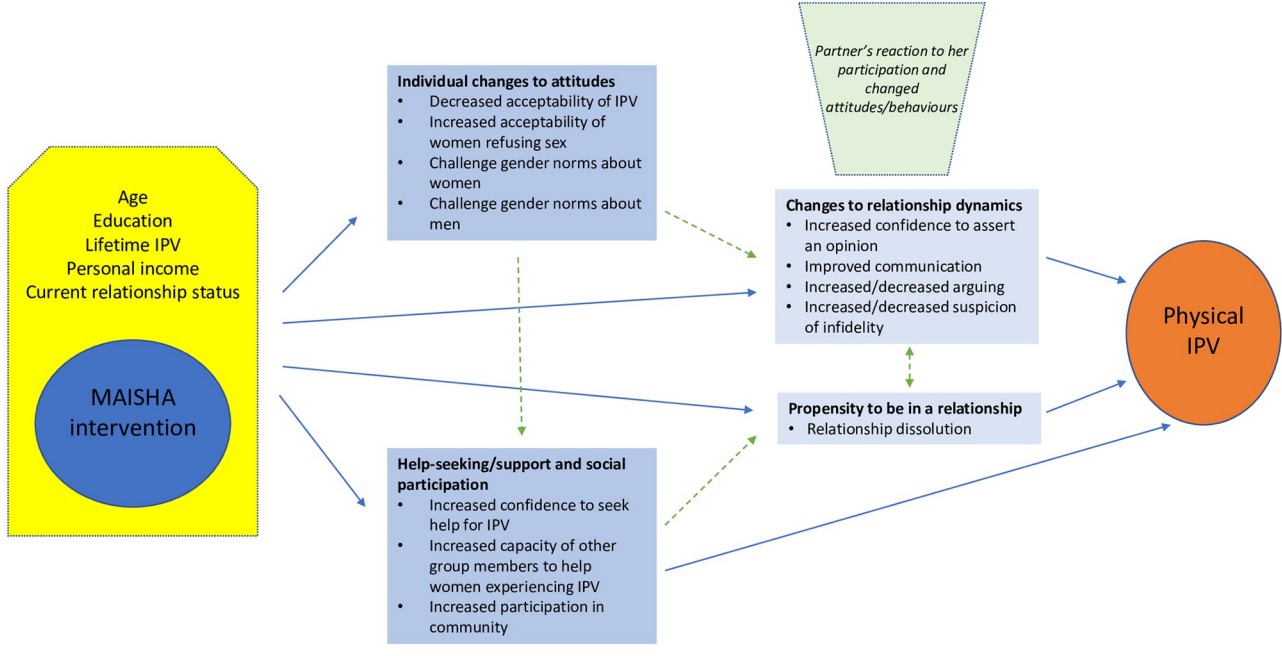

**Fig 4. Conceptual framework—MAISHA pathways of impact on physical IPV.**

## Outcome

The outcome for this analysis is women's reported experience of past year physical IPV at follow-up. This was measured among all ever-partnered women, including women who had (had) intimate partners with whom they did not cohabit. Details of questionnaire items used to construct this outcome are presented in Table 1.

## Potential effect modifiers

We hypothesised that intervention impact on physical IPV might vary according to the participant's age, her education, whether she had already experienced physical IPV at the outset of the study (i.e. preventing new onset versus promoting cessation of IPV), her personal income/financial independence (could look after the family on her income alone), and her current relationship status (all measured at baseline). These factors could plausibly affect her motivation to attend the intervention and engage with its content, her understanding of the content,

**Table 1. Questions used to construct physical IPV outcome.**

| Outcome indicator | How measured |
| --- | --- |
| Past year physical IPV (Yes; No) | CODED AS 'YES' IF: <br> Answers 'yes' to <u>at least one</u> of the following, in relation to the past 12 months: <br> Has your current partner or any other partner… <br> • Slapped you or thrown something at you that could hurt you? <br> • Pushed you or shoved you, or pulled your hair? <br> • Hit you with his fist or something else that could hurt you? <br> • Kicked you, dragged you or beaten you up? <br> • Choked or burnt you on purpose? <br> • Threatened to use or actually used a gun, knife or other weapon against you? <br> CODED AS 'NO' IF RESPONDS 'NO' TO <u>ALL</u> |

and her capacity to enact individual-level and relationship-level change in response to her involvement with MAISHA.

## Potential mediators

Potential mediators were grouped into four categories: 1) Woman's attitudes (acceptability of IPV, gender norms about women, gender norms about men); 2) Help-seeking and social capital (woman's confidence to seek help for IPV, participation in community meetings, capacity of other group members to help women experiencing IPV); 3) Relationship dynamics (woman's confidence within the relationship, communication with partner, arguments with partner, partner's suspicion that she is unfaithful); 4) Woman's relationship status (relationship dissolution among those in a relationship at baseline). Details of these variables are presented in S1 Table. The hypothesised interrelationships between these potential mediators of intervention effect on physical IPV are presented in Fig 2.

Mediator variables were measured at follow-up. The variables in the 'relationship dynamics' category pertained only to women who reported having had a partner in the past 12 months, relationship dissolution was relevant only to those women with a partner at baseline, while the other categories of mediator were relevant to all women.

We hypothesised that the direction in which 'relationship dynamics' changed would be influenced not only by the intervention itself and antecedent changes in the woman's attitudes/social participation, but also by her partner's reaction to her involvement in the intervention. Relationship dynamics could thus change in a positive direction (as intended), or negative direction if new tensions were introduced into the relationship as a result of her intervention attendance or changed attitudes and behaviours.

## Ethics and safety

Both trials were conducted in accordance with WHO recommendations on researching VAW [28]. Interviewers underwent extensive training on conducting VAW research, including the importance of maintaining privacy and confidentiality at all times, and minimising distress to respondents. Information about local support services was provided to all participants irrespective of whether they reported experiencing violence. Furthermore, for those women who reported experiencing violence, a referral system was in place to help access appropriate services and support. The trials were approved by the Tanzanian National Health Research Ethics Committee of the National Institute for Medical Research (reference NIMR/HQ/R.8a/Vol IX/1512), and the London School of Hygiene & Tropical Medicine research ethics committee (reference 11642).

## Statistical analysis

Baseline and follow-up data were recorded directly onto tablet computers with in-built checks to minimise missing or erroneous data. Data analysis was performed using STATA V.17. The data analysis involved three steps.

First, we performed sub-group analyses, separately for CRT01 and CRT02, to explore whether intervention impact on past year physical IPV differed by age (<35 years, 35+ years), education (primary or below, above primary), respondent's income (below median, above median), financial independence (couldn't look after family on her income alone, could look after family on her income alone), baseline marital status, and baseline experience of physical IPV (never, ever). We explored each potential effect modifier in separate logistic regression models, with random intercepts for group (unit of randomisation) to account for the clustered nature of the data. We estimated intervention effect (adjusted OR with 95% CI) on physical

IPV in each of the sub-groups by including an interaction term between the intervention and respective potential effect modifier. Additional adjustment was made for age (modelled as a linear effect), education (secondary/higher vs primary/none) and baseline physical IPV, where these were not already included in the model as a potential effect modifier. To test for the statistical significance of any observed differences in impact between sub-groups, we used the likelihood ratio test (LRT) to compare models with and without the interaction term.

The second stage of analysis estimated intervention impacts on potential mediators of intervention effect on physical IPV, separately for CRT01 and CRT02. Each potential mediator was considered separately. Adjusted OR (with 95% CI) for intervention impact on each potential mediator was estimated using a logistic regression model with random intercepts for group (unit of randomisation), and fixed effects terms for age (modelled as a linear effect), education (secondary/higher vs primary/none), and baseline measure of the respective mediator.

Next, we explored the associations between each of these potential mediators and past year experience of physical IPV at follow-up (separately for CRT01 and CRT02). Adjusted ORs with 95% CI were estimated in a mixed effects logistic regression model with physical IPV as the dependent variable, the respective mediator as an independent variable, and potential confounders of the association (age, education, current partnership status and baseline experience of physical IPV) included as covariates. Interactions between intervention arm and the mediator (IPV risk factor) of interest were also checked to assess if mediators were related to IPV similarly across intervention arms. As no consistent patterns of interaction were found, overall results for both arms are reported.

Finally, for CRT01 only, we modelled intervention effect on past year physical IPV, adjusting for each potential mediator separately. We examined the extent to which each potential mediator's inclusion in the basic model (including age, education and baseline physical IPV) attenuated intervention impact on IPV—interpreting greater attenuation as suggestive of the increased importance of that variable as a mediator of intervention impact on IPV. All analyses of intervention impact were conducted on an intention to treat basis.

## Results

Response rates were high in both trials, with 89% of participants in each completing both baseline and follow-up interview. Mean group size was also similar in both trials, with a mean of 19.7 (SD = 5.03) women per group in CRT01 and 19.2 (SD = 4.7) in CRT02. Attendance was higher in CRT02 compared to CRT01 (82% attended 7 or more sessions in CRT02, versus 67% in CRT01). In both trials, intervention and control arm women were similar with respect to a range of baseline characteristics, as reported in the main trial papers [18, 19] (S2 Table). Although control arm women in CRT01 had slightly higher levels of education and lower levels of sexual IPV than intervention arm women, these imbalances were adjusted for in relevant analyses of intervention impact.

Characteristics of women in the two trials are presented in Table 2. On average, women in CRT01 were older than women in CRT02 (mean age 39.6 years versus 33.1 years). They were slightly less likely to be currently married/living with a man as if married (73% versus 81%), reflecting greater levels of separation and widowhood. Women in CRT01 were less likely than women in CRT02 to have no/incomplete primary education, though only a fifth of women in either trial were likely to have progressed beyond primary level education. Across both trials, the vast majority of women (95%) had at least one child for whom they were responsible.

Several indicators suggest that women in CRT01 had more financial resources and independence than women in CRT02. They were more likely to have personally earned money in the past 12 months (97% versus 80%), and to have a higher income if they worked. 34% of

**Table 2. Baseline individual- and relationship-level characteristics of women in the MAISHA trials, presented separately for women in CRT01 and CRT02.**

| | CRT01 N = 919 | CRT02 N = 1125 |
|---|---|---|
| **Demographics** | | |
| Age (yrs) | | |
| *Mean (sd), [range]* | 39.6 (9.5), [19–70] | 33.1 (8.1), [18–50] |
| Currently married/living with a man as if married | 675 (73%) | 906 (81%) |
| Education | | |
| *None/incomplete primary* | 131 (14%) | 214 (19%) |
| *Completed primary* | 591 (64%) | 680 (60%) |
| *Attended secondary or above* | 197 (21%) | 231 (21%) |
| Number of children (<18 yrs) | | |
| *None* | 59 (6%) | 52 (5%) |
| *1–2* | 301 (33%) | 420 (37%) |
| *3+* | 559 (61%) | 653 (58%) |
| **Economic situation** | | |
| Personally earned money in past 12 months | 891 (97%) | 903 (80%) |
| Respondent's monthly income (TZA shillings) | | |
| *Median (IQR)* | 220,000 (110,000 to 440,000) | 105,000 (55,000 to 176,000) |
| Respondent could definitely look after family on her income alone (among those with income) | 306/891 (34%) | 183/894 (20%) |
| Experienced household-level financial hardship in past year* | 429 (47%) | 702 (62%) |
| **Gender and IPV Attitudes** | | |
| Attitudes accepting of IPV | 504 (55%) | 615 (55%) |
| Believes a woman is obliged to have sex with her husband even if she doesn't want to | 229 (25%) | 253 (22%) |
| Believes a woman should obey her husband's wishes even if she disagrees with them | 323 (35%) | 422 (38%) |
| Believes it must be the man who is primary provider for the family | 645 (70%) | 770 (68%) |
| **Past year experience of IPV (among ever-partnered women)** | | |
| Physical IPV | 172 (19%) | 281 (25%) |
| Sexual IPV | 151 (16%) | 235 (21%) |
| Emotional IPV | 366 (40%) | 501 (45%) |
| Economic abuse | 306 (38%) | 441 (43%) |

*As measured by past year experience of worry/stress about financial situation (a few or many times), plus forgoing or difficulty meeting costs of food/other necessities, rent/other bills, healthcare, or school fees/uniform/educational supplies.

women in CRT01 reported that they would be able to look after their family on their (the woman's) income alone, versus 20% of women in CRT02. Their households too were better off, with CRT01 women less likely than CRT02 women to report that their households had experienced financial hardship in the past year (47% versus 62%).

Though baseline attitudes on gender roles and the acceptability of IPV did not vary between women in the two trials, women in CRT01 were less likely than women in CRT02 to have experienced physical, sexual, emotional or economic IPV in the past 12 months. Further analysis shows this was largely accounted for by their older age (S3 Table).

## Subgroup analyses

The subgroup analyses of intervention effects among CRT01 women suggest that intervention impacts on physical IPV differed according to study participants' characteristics (Table 3). Effects appeared greater in women under 35 years than in women 35 years or older. They were also greater among people who had never experienced physical IPV at baseline (preventing new onset of IPV) than they were among those with a history of physical IPV at baseline (preventing continuation).

**Table 3. Adjusted odds ratios of intervention impact on past year physical IPV among different sub-groups of women in CRT01.**

| | Prevalence of past year IPV at follow-up n/N (%) | | Intervention impact on past year physical IPV | Likelihood Ratio Test[±] p-value |
|---|---|---|---|---|
| | Intervention arm | Control arm | aOR* (95%CI) | |
| | n/N (%) | n/N (%) | | |
| Overall study population | 68/485 (14%) | 82/434 (19%) | 0.63 (0.41–0.98) | |
| Baseline report of lifetime physical IPV | | | | p = 0.155 |
| Never physical IPV | 12/224 (5%) | 25/208 (12%) | 0.40 (0.19–0.86) | |
| Ever physical IPV | 56/261 (21%) | 57/226 (25%) | 0.75 (0.45–1.23) | |
| Age | | | | p = 0.136 |
| Under 35 years | 30/155 (19%) | 39/123 (32%) | 0.46 (0.25–0.86) | |
| 35+ years | 38/330 (12%) | 43/311 (14%) | 0.82 (0.49–1.39) | |
| Education | | | | p = 0.441 |
| Primary or below | 55/406 (14%) | 59/316 (19%) | 0.59 (0.36–0.94) | |
| Above primary | 13/79 (16%) | 23/118 (19%) | 0.84 (0.37–1.92) | |
| Income | | | | p = 0.361 |
| Below median income | 49/310 (16%) | 55/283 (19%) | 0.72 (0.44–1.19) | |
| Above median income | 19/175 (11%) | 27/151 (18%) | 0.49 (0.25–0.99) | |
| Financial independence | | | | p = 0.334 |
| Probably/definitely couldn't look after family on her income alone | 15/74 (20%) | 15/64 (23%) | 0.94 (0.38–2.30) | |
| Probably/definitely could look after family on her income alone | 53/411 (13%) | 67/370 (18%) | 0.58 (0.36–0.93) | |
| Baseline marital status | | | | p = 0.859 |
| Separated/divorced/widowed/ never married | 10/132 (8%) | 12/112 (11%) | 0.69 (0.27–1.78) | |
| Currently married/living as married | 58/353 (16%) | 70/322 (22%) | 0.63 (0.40–1.00) | |

*Estimated from mixed effects logistic regression models with random intercepts for MF group. Models included interaction term between intervention and sub-group characteristic, and fixed effects terms for age (linear), education (above primary/primary or below) and baseline past year physical IPV.

[±]Likelihood ratio test comparing model with and without interaction

Though evidence from the LRT tests of interaction was weak, the point estimates of adjusted odds ratios suggest that effects may also have been stronger among women with above median income, and those who could look after their family on their income alone.

There was no evidence of any interactions in CRT02, with no intervention effects on physical IPV observed in the overall sample or any of the subgroups (S4 Table).

## Intervention impacts on potential mediators

**Woman's attitudes and behaviours.** *CRT01*. The intervention was associated with reductions in the acceptability of IPV and the belief that a woman should be obliged to have sex with her husband (Table 4). Of these, attitudes accepting of IPV were strongly associated with increased risk of IPV (Fig 5), making reduced acceptability of IPV a potential mediator of intervention effect on physical IPV. There was less evidence that the intervention was associated with changes to broader gender attitudes around a woman's obligation to obey her husband or a man's role as primary provider for the family.

**Table 4. Intervention impact on potential mediators (at follow-up), presented separately for CRT01 and CRT02.**

| | CRT01 | | | CRT02 | | |
|---|---|---|---|---|---|---|
| | Intervention | Control | aOR* (95%CI) | Intervention | Control | aOR* (95%CI) |
| **Individual-level attitudes (among all ever-partnered women)** | n = 485 | n = 434 | | n = 550 | n = 575 | |
| Attitudes accepting of IPV | 215 (44%) | 243 (56%) | 0.45 (0.34–0.61) | 282 (51%) | 373 (65%) | 0.49 (0.36–0.66) |
| Believes a woman should be obliged to have sex with her husband | 67 (14%) | 78 (18%) | 0.66 (0.45–0.95) | 56 (10%) | 97 (17%) | 0.53 (0.36–0.80) |
| Believes a woman should obey her husband | 116 (24%) | 118 (27%) | 0.78 (0.56–1.08) | 131 (24%) | 199 (35%) | 0.56 (0.42–0.74) |
| Believes the man must be primary provider for the family | 295 (61%) | 290 (67%) | 0.72 (0.52–1.01) | 310 (56%) | 408 (71%) | 0.46 (0.33–0.63) |
| **Social participation/ help-seeking/ bystander action (among all ever-partnered women)** | n = 485 | n = 434 | | n = 550 | n = 575 | |
| Participation in community meetings | 170 (35%) | 136 (31%) | 1.20 (0.85–1.69) | 108 (20%) | 106 (18%) | 1.10 (0.78–1.55) |
| Very comfortable seeking help for IPV (hypothetical) | 292 (60%) | 229 (53%) | 1.29 (0.99–1.69) | 288 (52%) | 265 (46%) | 1.27 (0.95–1.70) |
| Group level confidence to intervene in cases of IPV (hypothetical) | 0.70 (0.142) | 0.62 (0.125) | β = 0.088 (0.026–0.149) | 0.53 (0.179) | 0.42 (0.124) | β = 0.103 (0.025–0.181) |
| **Relationship dynamics (past year) (among women partnered in past year)** | n = 405 | n = 368 | | n = 502 | n = 511 | |
| Confident to assert an opinion different to partner's | 306 (76%) | 245 (67%) | 1.63 (1.18–2.24) | 320 (64%) | 293 (57%) | 1.33 (0.95–1.85) |
| Good communication with partner | 229 (57%) | 213 (58%) | 0.97 (0.69–1.35) | 226 (45%) | 235 (46%) | 0.98 (0.73–1.32) |
| Partner often suspicious that she is unfaithful | 85 (21%) | 59 (16%) | 1.28 (0.81–2.02) | 102 (20%) | 90 (18%) | 1.13 (0.78–1.64) |
| Argued with partner over her not fulfilling her role as wife and mother | 126 (31%) | 108 (29%) | 1.06 (0.74–1.52) | 154 (31%) | 126 (25%) | 1.36 (1.01–1.82) |
| Argued with partner over his inability/unwillingness to provide for the family | 162 (40%) | 151 (41%) | 0.93 (0.69–1.24) | 188 (37%) | 183 (36%) | 1.07 (0.83–1.38) |
| Argued with partner over her disobeying/disrespecting him | 69 (17%) | 73 (20%) | 0.82 (0.57–1.19) | 79 (16%) | 64 (13%) | 1.31 (0.92–1.88) |
| Argued with partner over him treating her/ her children disrespectfully | 74 (18%) | 77 (21%) | 0.83 (0.58–1.20) | 99 (20%) | 83 (16%) | 1.26 (0.92–1.75) |
| Argued with partner (any topic) | 268 (66%) | 259 (70%) | 0.78 (0.56–1.09) | 336 (67%) | 348 (68%) | 0.99 (0.75–1.29) |
| **Relationship dissolution (among those partnered in past year at baseline)** | n = 426 | n = 379 | | n = 503 | n = 527 | |
| Left partner since baseline | 64 (15%) | 51 (13%) | 1.20 (0.73–1.96) | 74 (15%) | 73 (14%) | 1.08 (0.75–1.54) |

*Odds ratios estimated using logistic regression models with random intercepts for MF group, and fixed terms for age (linear term), education (secondary/higher versus primary/none), and baseline measure of the respective mediator.

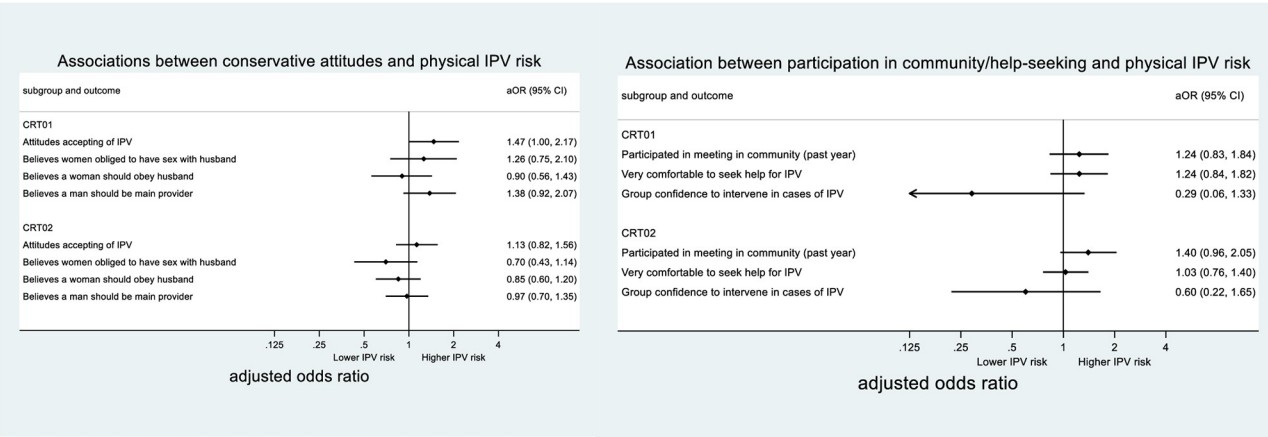

**Fig 5. Association between individual- and group-level potential mediators and physical IPV risk.**

In keeping with these results, the addition of attitudes accepting of IPV to the model of intervention impact on physical IPV led to a slight attenuation of effect (11% reduction in aOR), while the addition of the other attitudes did not do so (Table 5).

*CRT02.* All attitudes relating to IPV and gender roles were impacted on in a progressive direction in the intervention group compared to the control group (Table 4). However, none of these attitudes were strongly associated with IPV risk (Fig 5).

**Help-seeking/support to women experiencing IPV/social participation.**   *CRT01.* Women in the intervention arm were more likely to say they would feel very comfortable seeking help if they themselves were experiencing IPV, and a higher percentage of people in their microfinance groups said that they would feel confident to intervene if someone they knew were experiencing IPV (Table 4). Only the latter, group confidence to intervene, was associated with lower risk of IPV for the respondent (Fig 5). Though this association was not statistically significant, when group confidence was added to the model of intervention impact on physical IPV there was a sizeable attenuation of effect (16% reduction in aOR) (Table 5).

The intervention does not appear to have impacted on women's participation in meetings in the community (Table 4), which anyway was not associated with risk of physical IPV (Fig 5).

*CRT02.* Impacts on help seeking and group confidence to intervene were also observed in CRT02, though from a baseline of slightly lower confidence to seek help/support others. Again, only group confidence to intervene was associated with lower IPV risk, but as in CRT01 the association was not statistically significant (Table 4).

As with CRT01, the intervention did not impact on women's levels of participation in meetings in the community, with lower overall participation among women in CRT02 compared to CRT01 (Table 4). Interestingly, there was some suggestion that this indicator was anyway associated with increased risk of IPV in this population (though the confidence interval included unity). Therefore, even if participation in meetings had increased in the intervention arm it would likely not have led to a decrease (and may even have led to an increase) in IPV risk among these women.

**Relationship dynamics.**   *CRT01.* The intervention was not associated with improvements in communication with the partner in CRT01, nor was it associated with an increase or decrease in arguing with the partner over a variety of topics related to gender roles/perceived

**Table 5. Intervention impact on women's past year experience of physical IPV, with and without adjustment for potential mediators—CRT01.**

| Mediator adjusted for: | CRT01 | |
|---|---|---|
| | aOR* (95%CI) for MAISHA impact on past year physical IPV | % change in aOR after addition of mediator |
| **Whole sample of ever-partnered women** | n = 919 | |
| Model without mediators | 0.63 (0.41–0.98) | - |
| *Individual-level attitudes* | | |
| Acceptability of IPV | 0.67 (0.43–1.03) | 11% |
| Believes a woman should be obliged to have sex with her husband | 0.64 (0.42–0.98) | 3% |
| Believes a woman should obey her husband | 0.63 (0.41–0.98) | 0% |
| Believes the man must be primary provider for the family | 0.64 (0.42–0.99) | 3% |
| *Help-seeking/social participation/bystander intentions* | | |
| Participation in community meetings | 0.63 (0.41–0.97) | 0% |
| Very comfortable seeking help for IPV (hypothetical) | 0.63 (0.41–0.97) | 0% |
| Group level confidence to intervene in cases of IPV (hypothetical) | 0.69 (0.45–1.07) | 16% |
| **Sub-group in relationship in past year** | n = 773 | |
| Model without mediators | 0.67 (0.43–1.02) | - |
| *Relationship dynamics (past year)* | | |
| Confident to assert an opinion different to partner's | 0.67 (0.43–1.02) | 0% |
| Good communication with partner | 0.67 (0.43–1.04) | 0% |
| Partner often suspicious that she is unfaithful | 0.57 (0.35–0.93) | -30% |
| Argued with partner over her not fulfilling her role as wife and mother | 0.65 (0.42–1.00) | -6% |
| Argued with partner over his inability/ unwillingness to provide for the family | 0.66 (0.43–1.02) | -3% |
| Argued with partner over her disobeying/ disrespecting him | 0.68 (0.45––1.04) | 3% |
| Argued with partner over him treating her/ her children disrespectfully | 0.68 (0.45–1.02) | 3% |
| Argued with partner (any topic) | 0.70 (0.45–1.06) | 9% |
| **Subgroup with partner at baseline** | n = 805 | |
| Model without mediators | 0.63 (0.41–0.98) | - |
| *Relationship dissolution* | | |
| Left partner since baseline | 0.64 (0.41–0.98) | 3% |

*Odds ratios estimated using logistic regression models with random intercepts for MF group, and fixed terms for age (linear term), education (secondary/higher versus primary/none), baseline past year physical IPV, and follow-up measure of the respective mediator (where applicable).

transgression of gender roles (Table 4). Better communication is, however, associated with lower IPV risk, and arguing over gender roles associated with increased IPV risk, suggesting that intervention efficacy in preventing IPV could be enhanced if greater improvements in communication within the relationship were achieved. The intervention was associated with a slight (not statistically-significant) reduction in 'any arguing', and when this mediator was

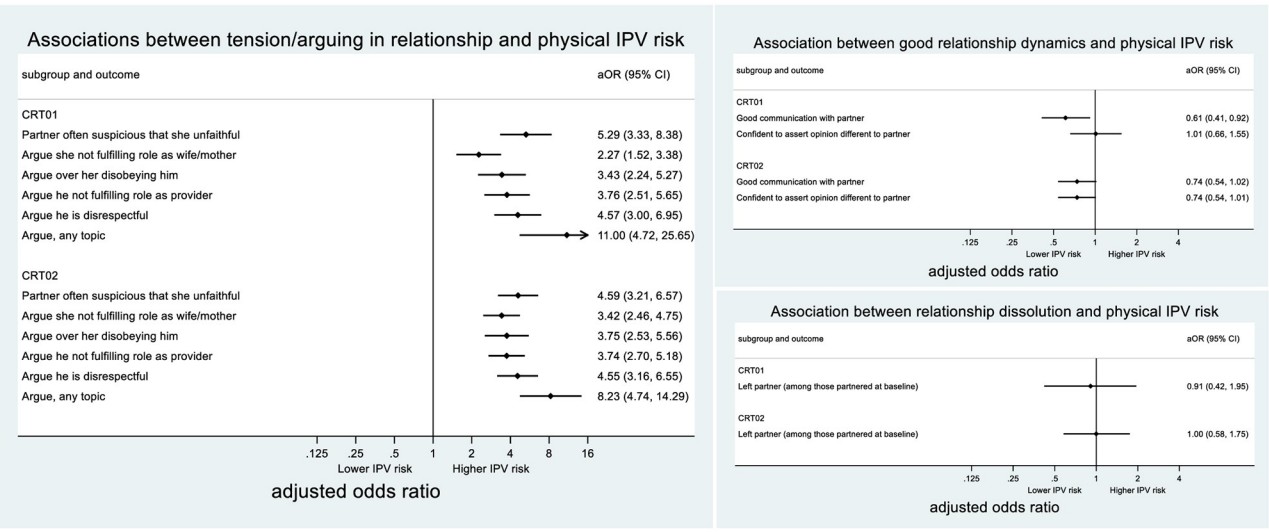

**Fig 6. Association between relationship-level mediators and physical IPV risk.**

included in the model of intervention effect on IPV, there was a small attenuation in effect size (9% reduction in aOR) (Table 5).

Women in the intervention group were more likely than their control counterparts to be confident to assert an opinion different to their partner's (Table 4), but this confidence was not associated with decreased IPV risk (Fig 6).

Women in the intervention arm were slightly more likely to report that their partner was often suspicious that they were unfaithful, though this difference was not statistically significant (Table 4). In turn, greater suspicion was associated with increased risk of IPV (Fig 6). When partner suspicion was accounted for in the model of intervention effect on physical IPV, the effect size *increased* suggesting that intervention impacts on IPV would have been greater in the absence of increased partner suspicion (Table 5).

*CRT02.* In CRT02, the intervention was similarly not associated with general improvements in communication with the partner. It was, however, associated with increased arguing about the woman not fulfilling her role, and weakly associated with increased arguing about the woman disobeying her partner and the man being disrespectful (Table 4). From a baseline of less frequent arguing about gender roles in CRT02, the intervention appears to have increased arguing among intervention arm women to similar levels to those observed in CRT01 (Table 4). In turn, arguing was strongly associated with increased IPV risk (Fig 6).

There was less evidence that women in the intervention group were more confident than control women to assert an opinion different to their partner's (Table 4) (an indicator which was associated with reduced IPV risk in this trial) (Fig 6).

There was no strong evidence of intervention impact in either direction on partner's suspicion that the woman was unfaithful.

**Separation.** *CRT01.* In CRT01, there was no evidence that women in the intervention arm, partnered at baseline, were more likely than women in the control arm to separate from their partner during the course of the trial (an indicator anyway not associated with IPV risk).

*CRT02.* In CRT02, there was similarly no association observed between the intervention and relationship dissolution (Table 4). There was also no suggestion that separation was associated in either direction with IPV risk (Fig 6).

## Discussion

The two MAISHA trials have offered us insights into the *contexts in* and *mechanisms through* which a gender training intervention can work to prevent IPV, as well as potential barriers to the effectiveness of such interventions. Among women receiving microfinance (CRT01), the MAISHA intervention reduced women's experience of physical IPV, with greater impacts observed among younger women (<35 years versus 35+), those in non-violent relationships at baseline, and (though evidence was weaker) among women with greater financial independence. In CRT02, however, despite higher levels of attendance to the intervention, no impacts were seen on physical IPV either overall or within specific sub-groups. While positive impacts were seen on individual- and group-level potential mediators in both trials—reduced acceptability of IPV, confidence to seek help if experiencing IPV (hypothetical scenario), and group-level confidence to intervene in cases of IPV—relationship-level mediators such as communication within the relationship do not appear to have been positively impacted on in either trial. Indeed, in CRT02 it appears that arguments with partners over perceived transgressions of gender roles by women may actually have increased. Evidence also suggests that reduced acceptability of IPV may have been less strongly linked to reduced IPV risk among women in CRT02 than it was among women in CRT01. The intervention did not appear to lead to increased rates of separation in either trial.

The differential impacts seen among different sub-groups of women in CRT01 are instructive for the potential targeting of these types of interventions and the identification of potential barriers to intervention effectiveness. In CRT01, we observed greater impacts among younger women compared to older women, and among women with no prior experience of physical IPV at baseline compared to those who had already experienced physical IPV at baseline. These findings suggest that the intervention was more effective at preventing the new onset of physical IPV where it wasn't previously occurring, than it was at preventing the continuation of violence in already violent relationships. It is perhaps logical that an intervention targeted to women only might be more effective at preventing women from entering into violent/less gender-equitable relationships, or at fostering women's communication and negotiating skills that prevent currently non-violent relationships from escalating into conflict, than it is at changing the nature of relationships in which a man's use of violence against his partner is already an established dynamic.

Findings from the sub-group analysis may also help explain why we didn't observe impacts on physical IPV in CRT02. Although women in CRT02 were on average younger than women in CRT01, and we might therefore expect greater impacts in CRT02 than those observed in CRT01, women in CRT02 were also more likely to be experiencing IPV at baseline. This is the subgroup for whom the intervention was demonstrably less effective in CRT01.

In CRT01, we also observed stronger impacts among women with greater financial independence at baseline, in line with arguments that structural factors such as poverty and women's financial dependence on men can trap women in relationships and act as barriers to the success of violence prevention initiatives [5, 29, 30]. Qualitative evidence from women enrolled in CRT01 also points towards synergies between the gender training and microfinance, with women discussing the importance of their financial security and independence in allowing them to stand up for themselves [27]. Women in CRT02 were less likely than women in CRT01 to have their own income, and those who did earned on average considerably less than their counterparts in CRT01. They were also less likely to report that they could look after the family on their income alone. It is important to note that it is not possible on the basis of our data to infer whether women in CRT01 were in a stronger economic position as a result of their participation in microfinance, or whether their participation in microfinance was driven

or facilitated by greater pre-existing levels of financial resources/independence. Indeed, it is likely a bidirectional association. Regardless of the reason for this difference, it is plausible that greater poverty and/or lack of financial autonomy prevented women in CRT02 from using their MAISHA training to enact change at the relationship-level.

In both trials, the intervention impacted on several factors that we hypothesised to be potential pathways to IPV prevention. In particular, there were large impacts on attitudes. In both CRT01 and CRT02, the intervention was associated with significant reductions in the acceptability of IPV. However, these attitudes were not uniformly related to IPV risk in the two trials: Non-acceptability of IPV was linked to lower IPV risk in CRT01, thereby making it a potential mediator of intervention effect in CRT01; but not in CRT02, indicating that attitudinal change may not be sufficient to bring about reduced risk of IPV in this population of women. Since a woman's attitudes can only impact indirectly on her own risk of experiencing IPV—through her choice of partner, her decisions on whether or not to remain with an abusive partner, or her motivation to negotiate revised relationship parameters and dynamics with her partner—it is not surprising that the association between attitudes and IPV might be modified by context. It is plausible that a shift to more progressive attitudes can only translate into changes in IPV risk, if a woman also has sufficient bargaining power within the relationship, engendered by other factors such as access to and autonomy over financial resources.

The intervention also impacted positively on broader gender attitudes in both trials, with even larger impacts observed in CRT02 than CRT01. These attitudes, however, were less strongly linked to reduced IPV risk in CRT01, than were attitudes explicitly related to the acceptability of IPV. They were also unrelated to IPV risk in CRT02. These findings are in line with those from the SASA! study, a trial of a community mobilisation VAW prevention intervention in Uganda, which found that attitudes on the acceptability of IPV were more influential mediators of intervention effect than were broader gender attitudes [31]. They support the idea that broader transformations in gender norms and attitudes may not be sufficient in themselves to prevent violence, if specific attitudes around the acceptability of violence against women are not also directly addressed and challenged. The finding that attitudes towards a woman's right to refuse sex are not strongly linked to physical IPV risk, also suggests that sexual coercion within intimate relationships, though often reinforced by physical violence, is a distinct phenomenon that transcends relationships with and without physical violence [32–34].

In both trials, there was a weak association between the intervention and women's reported comfort seeking help in the hypothetical scenario that they were experiencing IPV. However, comfort in seeking help was not associated with reduced IPV risk in either trial. It is possible that a bi-directional relationship between comfort seeking help and IPV risk may have led to this net zero association. Though help seeking might reduce IPV risk in the long term, those with experience of IPV might be more familiar and comfortable with the process of seeking help and respond accordingly in relation to the hypothetical scenario. Conversely, women with no experience of IPV might envisage more reticence in telling others. If this is the case, we may have underestimated the potential role that increased comfort seeking help might have as a mediator of longer-term intervention effect on IPV.

Confidence to intervene in the hypothetical case that another woman was experiencing IPV also increased, with higher group-level percentages of women confident to intervene in intervention groups compared to control groups (8.8% higher in intervention versus control groups in CRT01; 10.3% higher in intervention versus control groups in CRT02). Group-level confidence to intervene was associated with reduced IPV risk in both trials (not statistically significant), and when added to the model of intervention effect on physical IPV in CRT01 led to a moderate attenuation of effect. This evidence, in accordance with an evaluation of the scaled-

up IMAGE intervention in South Africa [35], is suggestive of the role that improved social support networks may have in mediating intervention effect on individual women's experiences of physical IPV.

Participation in community meetings did not increase in either trial. There was also evidence that participation in meetings was associated with increased rather than decreased IPV risk, particularly among control arm women in CRT02. This suggests that interventions with men and others in the community are necessary in order to increase partner- and community-level acceptance of women taking on more active roles outside of the house and within the community [36].

Though good communication between a woman and her partner, as indicated by their discussing each other's daily activities and worries, was associated with reduced IPV risk in both trials, there was no evidence from either trial of improved communication among intervention arm women. This runs counter to qualitative evidence in which women who had participated in the intervention cited more positive communication with their partners and better conflict resolution skills [27]. It is possible that the lack of impact on communication observed here is artefact of the crude measure of communication used in this analysis (based on topic and frequency rather than quality of discussion). Interestingly, women's confidence to assert an opinion different to their partner's did increase, particularly in CRT01, in line with findings from the qualitative research in which women talked about greater confidence to express themselves about their own perceptions of situations and dynamics within the relationship that they wished to change [27]. Increased confidence was in turn associated with reduced IPV risk in CRT02 but not in CRT01.

Arguments between a woman and her partner were a strong risk factor for IPV in both trial populations. In neither trial did the intervention lead to a decrease in arguing, and in CRT02 there was some suggestion that arguments over perceived transgression of gender roles by the woman may actually have increased among women in the intervention arm. It is plausible that individual-level changes to a woman's attitudes, confidence and behaviours—in the absence of any intervention with the male partners—has led to increased tension in relationships that from the outset were characterised by greater gender inequality [37]. This, in turn, could have cancelled out the effect of other pathways we might have expected to lead to reduced IPV. The importance of involving men in IPV prevention programming, including through couples-based interventions, has gained traction in recent years, and CRTs of several such interventions have demonstrated their effectiveness in preventing IPV [25, 26, 38–40]. Indashyikirwa, for example, a 21-session curriculum delivered to couples recruited from Village Savings and Loans Associations in Rwanda, led to substantial reductions in women's reported experience of physical and/or sexual IPV, as well as reduced relationship conflict and improved communication more broadly [25].

In CRT01, the data also point to a possible increase in women reporting that their partners often suspected them of infidelity. In turn, the mediation analysis suggests that intervention impacts on physical IPV may have been even greater in the absence of this suspicion. Again, this unintended consequence (if real and not a spurious finding) could plausibly arise from the lack of any intervention with male partners—with greater intervention impacts possible if men's attitudes linking infidelity to the acceptability of women refusing sex or seeking work outside the home were addressed. Indeed, in the SASA! trial of a community intervention that involved entire communities—both women and men—to address community-level norms underpinning high levels of VAW, levels of male partners' suspicion over infidelity did fall. This fall came alongside progressive shifts in both women's and men's attitudes around IPV and women's right to refuse sex. The intervention led to community-level reductions in IPV, with changes in men's attitudes and reduced suspicion over infidelity emerging as potentially important mediators of this impact [31].

Finally, although we might hypothesise that interventions such as MAISHA could impact on violence by motivating and better equipping women to leave abusive relationships, MAISHA does not appear to have made women more likely to leave their partner in either trial. It is therefore through individual-level change and improvements to relationship dynamics, some of which have been described above and others of which have likely not been captured in this analysis, that MAISHA has reduced women's risk of physical IPV in CRT01. A similar pattern has been observed in relation to other IPV prevention interventions such as IMAGE in South Africa [13] and SASA! In Uganda [31].

## Strengths and limitations

This study has many strengths, not least that it comprises two comparably designed cluster randomised trials of the same intervention delivered in different contexts. This allows us not only to be confident in the internal validity of our intervention/control comparisons, but also to compare intervention impacts and mechanisms when implemented among different populations of women. In both trials, we achieved high levels of intervention attendance and high retention rates. We analysed impacts on both IPV and potential mediators following the intention to treat principle, thereby minimising selection bias. Baseline data enabled us to conduct a range of sub-group analyses as well as adjust for baseline imbalances between the trial arms.

Nevertheless, there are limitations to this analysis. As with other violence research studies, our analysis is of self-reported data which may be prone to respondent bias. To reduce risk of under-reporting of IPV, we used standardised questions that are widely used in violence research, administered by interviewers who had received extensive training on the conduct of VAW research. However, some indicators used in the mediation analysis, such as communication with partner, are crude relative to the complexity of lived relationship dynamics and may not capture the dynamics most relevant to IPV risk. The qualitative components of the MAISHA trials offer deeper insights into areas such as communication and power imbalances within the relationship [27].

While our data encompass a range of important characteristics relevant to intervention effectiveness, we were not able to explore all potential pathways of effect or effect modifiers. For example, too few women had entered into new partnerships during the course of the trials to explore the role that the intervention had on propensity to enter into a relationship, or on women's criteria for choosing a new partner. Similarly, very few women in our sample had no children, meaning we were unable to explore whether having children might modify women's capacity/willingness to enact change in their relationships after engaging with the intervention.

Another limitation is that the data on mediators and physical IPV are cross-sectional (from the same follow-up time-point), precluding inference about causality or the temporality of associations. The randomised design and intention to treat analysis, along with baseline data showing intervention and control groups to be highly comparable prior to intervention implementation, lend support to the interpretation that MAISHA positively impacted on the mediating variables. However, we cannot rule out a loop of causality in the latter end of the causal pathway, whereby the change in mediator is brought about by a reduction in IPV rather than the reverse scenario. Prior to conducting any analysis, we laid out theoretically and empirically grounded plausible pathways of effect in a conceptual framework. Our analysis and interpretation thus reflect our *a priori* assumptions about causality and direction of effect, rather than definitively testing a causal pathway.

Another limitation of our analysis is that we have examined the potential role of each mediator separately, when in all likelihood many occur in tandem with each other—hence

the role of each mediator may have been confounded by other potential mediators. Due to the complex process of change and multiple concurrent pathways involved in IPV prevention, our analysis attempts to identify which types of mediators *play a role in* reducing IPV rather than producing precise estimates of the proportion of intervention effect that can be attributed to each.

## Conclusions

In summary, this study provides important insights to the field of IPV prevention research. The analysis not only sheds light on potential pathways of effect through which the MAISHA intervention has impacted on violence in CRT01, and that future violence prevention interventions might exploit, but also identifies contextual factors which might impede intervention effect and act as barriers to success at different points along the prevention pathway. Importantly we find that the MAISHA intervention is more effective in younger women with no history of IPV at the outset of the intervention, and potentially more effective among women with higher levels of financial independence. This might in part explain the lack of impact on physical IPV in CRT02.

These findings point to the importance of addressing structural factors such as poverty and women's economic dependence on men that trap women in potentially abusive relationships. This might be achieved through incorporating economic strengthening components into IPV prevention interventions, or delivering IPV programming alongside pre-existing development activities which seek to improve women's access to and control over financial resources. It is important to note that enhanced intervention efficacy must be balanced against the risks sometimes inherent in microfinance based IPV prevention interventions, where attendance can be adversely affected by participants' difficulties keeping up with loan repayments and the demands of running a business [41, 42]. However, recent reviews have highlighted the effectiveness of IPV prevention interventions that combine economic strengthening (such as cash transfers, village savings and loans associations or microfinance) and gender transformative components [5, 6, 15]. As the global scale and reach of social protection schemes such as cash transfer programming increases from an already large base, this may provide expanding opportunities and platforms for scaled-up delivery of gender transformative IPV prevention interventions [15].

Finally, we see more impact on individual-level rather than relationship-level mediators in both trials, and attitudes less strongly linked to IPV risk in CRT02 compared to CRT01. Our findings also illustrate the potential for IPV prevention programmes to impact on specific relationship-level mediators in the unintended direction, for example increasing arguments between intimate partners. These perverse consequences with respect to specific mediators have the potential to counteract other positive effects of an intervention. An awareness of the potential for adverse consequences should be incorporated into intervention programming, and pre-emptive steps taken to ameliorate these risks. It should also inform larger strategic conversations around IPV prevention, such as the importance of involving and targeting men as well as women, including with couples-based and community-level interventions [6, 36]. This may be particularly important with respect to older women where (potentially violent) relationship dynamics may already be deeply entrenched. In order to shift such patterns, not only must male norms and behaviours be targeted, but community-level responses to VAW also strengthened. The SASA! community mobilisation intervention in Uganda, for example, through addressing risk and protective factors at multiple levels of the social ecology, was effective in promoting the cessation of violence within previously abusive relationships, in addition to preventing the new onset of abuse [31]. Through addressing these broader structural and

socio-ecological factors, we can more effectively empower women and enhance IPV prevention interventions.

## Supporting information

**S1 Checklist. CONSORT checklist.**
(DOCX)

**S1 Fig. MAISHA theory of change.**
(PDF)

**S1 Table. Questions used to construct potential mediator variables.**
(DOCX)

**S2 Table. Baseline characteristics of participants in CRT01 and CRT02, disaggregated by trial arm.**
(DOCX)

**S3 Table. Odds ratios comparing past year experience of IPV in CRT02 versus CRT01, before and after adjustment for respondent's age.**
(DOCX)

**S4 Table. Adjusted odds ratios of intervention impact on past year physical IPV among different sub-groups of women in CRT02.**
(DOCX)

**S1 Protocol. A cluster randomized controlled trial to assess the impact on intimate partner violence of a 10-session participatory gender training curriculum delivered to women taking part in a group-based microfinance loan scheme in Tanzania (MAISHA CRT01): Study protocol.**
(DOCX)

**S2 Protocol. A cluster randomised controlled trial to assess the impact on intimate partner violence of a 10-session participatory social empowerment intervention for women in Tanzania (MAISHA CRT02): Study protocol.**
(DOCX)

## Acknowledgments

We wish to thank the MAISHA study participants for their time and commitment. We also thank the MAISHA research team for their contribution and dedication to implementing the MAISHA study, and the administration teams at the Mwanza Intervention Trials Unit and London School of Hygiene and Tropical Medicine for their support.

## Author Contributions

**Conceptualization:** Tanya Abramsky, Diana Sanchez Guadarrama, Saidi Kapiga, Sheila Harvey.

**Formal analysis:** Tanya Abramsky.

**Funding acquisition:** Saidi Kapiga, Shelley Lees, Sheila Harvey.

**Investigation:** Tanya Abramsky, Saidi Kapiga, Grace Mtolela, Flora Madaha, Shelley Lees, Sheila Harvey.

**Methodology:** Tanya Abramsky, Diana Sanchez Guadarrama.

**Project administration:** Grace Mtolela, Flora Madaha.

**Writing – original draft:** Tanya Abramsky.

**Writing – review & editing:** Tanya Abramsky, Diana Sanchez Guadarrama, Saidi Kapiga, Grace Mtolela, Flora Madaha, Shelley Lees, Sheila Harvey.

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
