## [Decision Letter · Decision Letter 0]

26 Apr 2023

PGPH-D-22-01563

Pathways to reduced physical intimate partner violence among women in north-western Tanzania: Evidence from two cluster randomised trials of the MAISHA intervention

Dear Dr. Abramsky,

Thank you for submitting your manuscript to PLOS Global Public Health. After careful consideration, we feel that it has merit but does not fully meet PLOS Global Public Health’s publication criteria as it currently stands. Therefore, we invite you to submit a revised version of the manuscript that addresses the points raised during the review process.

Please address the reviewers' comments, including the need to demonstrate baseline balance in characteristics with statistical tests.

We look forward to receiving your revised manuscript.

Kind regards,

Tia M. Palermo

Academic Editor

Journal Requirements:

Additional Editor Comments (if provided):

Reviewers' comments:

Reviewer's Responses to Questions

**Comments to the Author**

1. Does this manuscript meet PLOS Global Public Health’s publication criteria? Is the manuscript technically sound, and do the data support the conclusions? The manuscript must describe methodologically and ethically rigorous research with conclusions that are appropriately drawn based on the data presented.

Reviewer #1: Yes

Reviewer #2: Partly

2. Has the statistical analysis been performed appropriately and rigorously?

Reviewer #1: Yes

Reviewer #2: No

3. Have the authors made all data underlying the findings in their manuscript fully available (please refer to the Data Availability Statement at the start of the manuscript PDF file)?

Reviewer #1: No

Reviewer #2: Yes

4. Is the manuscript presented in an intelligible fashion and written in standard English?

Reviewer #1: Yes

Reviewer #2: Yes

5. Review Comments to the Author

Reviewer #1: I congratulate the authors on a well-executed study and clearly written manuscript. The manuscript explored the pathways to reductions in physical IPV experience in two trials where the 10 session Maisha curricular was delivered i.e CRT01 involving women in existing micro-finance groups that were doing well with their loan repayments and CRT02 involving women recruited from the community and with no prior micro-finance group exposure. Positive impacts of intervention were reported in CRT01 and not CRT02. Sub-analyses showed that impacts in CRT01 were greater/significant among younger women who had no IPV exposure at baseline and no impacts recorded among women with IPV exposure at baseline or violent relationships. Mediator analyses implicated attitudinal changes i.e reduced acceptability of IPV as critical for observed reduced risk of physical IPV i. CRT01. Other anticipated mediator pathways were not established e.g improved communication was not associated with positive intervention impacts but those that had better communication at baseline had reduced physical IPV risk. Overall, the authors highlight the importance of structural intervention and women being less financially dependent in preventing the onset of IPV risk overall especially among younger women through low acceptability of IPV, decisions they make about partners and better communication that mitigates exacerbation of arguments to conflict and IPV. They also clearly identify the limitations of the study including the measurement of mediators cross-sectionally and the impossibility of establishing the directionality of some of the assumed associations/ pathways. I have few comments for the authors to consider attending to:

Background/Introduction

1. The manuscript is written in a style that assumes that the audience are familiar with the context of the study sites in Tanzania. However, my view is that it may be worthwhile to provide some context specific description to better understand some of the conceptualisation of pathways to change and intervention. For example, how does the context’s social organisation and culture relate to the fluidity of relationships or the stability of marriages or living together as married or the dynamics relevant to dissolving relationships.

Methods

2. For the mediator analysis, can the authors specify and reference how they used cutoffs to determine whether mediator effects were small, weak or large in the data analysis section.

3. Can the authors provide more detail about the criteria for selection of neighbourhood in CRT02 rather than just saying “suitable neigbourhoods”

Results

4. Line 304- Can the abbreviation LRT be written in full first time.

5. Table 1 presents disaggregated %s of different participant characteristics in CRT01 & CRT02. However, it does present data to indicate that tests for statistical significance of differences in proportions were conducted. Given the different sample sizes in the 2 trials, such tests can conclusively support whether the samples were statistically different. The text accompanying the table implies significant difference, but the results of statistical tests are needed as evidence of this.

Discussion

6. Given the reported impacts of intervention, it is important that the authors consider adding text that emphasises the limitation/ challenges of not working with male perpetrators of IPV to address IPV continuation among abused women. Currently the manuscript has emphasised that the VAW field prioritises economic empowerment programmes targeting women to reduce IPV. Yet the data presented also shows some unintended impacts among CRT01 women such as the backlash of increased arguments over gender roles and other conflicts. Ultimately CRT02 worked with women addressing their individual vulnerabilities to IPV but this was insufficient to lead to positive relational and IPV impacts without the economic empowerment and the non-involvement of male partners.

7. The authors refer to similar barriers observed with other interventions such as IMAGE and SASA. In my view it is warranted that the authors reflect and provide some answers/suggestions to questions such as “How should the field pivot in terms of interventions that address ongoing IPV and among older women?” What design or content issues require emphasis/ attention or inclusion” “How could IPV interventions with economic empowerment components have maximised impacts and brought to scale in LMIC settings?”

Reviewer #2: Please see attached file. I have detailed my comments in the attached file. The main comments are about the randomization. Other comments include sample construction and considering the endogeneity of the results.

6. PLOS authors have the option to publish the peer review history of their article (what does this mean?). If published, this will include your full peer review and any attached files.

**Do you want your identity to be public for this peer review?** For information about this choice, including consent withdrawal, please see our Privacy Policy.

Reviewer #1: No

Reviewer #2: No

---

## [Editor Report · Decision Letter 1]

25 Sep 2023

Pathways to reduced physical intimate partner violence among women in north-western Tanzania: Evidence from two cluster randomised trials of the MAISHA intervention

PGPH-D-22-01563R1

Dear Miss Abramsky,

We are pleased to inform you that your manuscript 'Pathways to reduced physical intimate partner violence among women in north-western Tanzania: Evidence from two cluster randomised trials of the MAISHA intervention' has been provisionally accepted for publication in PLOS Global Public Health.

Best regards,

Tia M. Palermo

Academic Editor
